# The Impact of Internalized Stigma on LGBT Parenting and the Importance of Health Care Structures: A Qualitative Study

**DOI:** 10.3390/ijerph18105373

**Published:** 2021-05-18

**Authors:** Carolina Alday-Mondaca, Siu Lay-Lisboa

**Affiliations:** Facultad de Humanidades, Escuela de Psicología, Universidad Católica del Norte, Antofagasta 1270709, Chile; slay@ucn.cl

**Keywords:** LGBTIQ+, maternity, paternity, parenting, internalized stigma, health care system

## Abstract

Research on LGBTIQ+ families has focused on the effects of being in a diverse family on the development of children. We seek to show the experience of parenthood from the perspective of LGBTIQ+ people, considering its particularities and the role that health care services play as a potential support network. We used the biographical method through open-ended interviews, participants were LGBT people, and key informants from Chile, Colombia, and Mexico were selected based on a sociostructural sampling. We found that internalized stigma impacts LGBTIQ+ parenting in five ways: the impossibility of thinking of oneself as a parent, fear of violating children’s rights, fear of passing on the stigma, fear of introducing their LGBTIQ+ partner, and the greater discrimination that trans and intersex people suffer. We identified gaps in health care perceptions: the need to guarantee universal access to health care, the need to include a gender perspective and inclusive treatment by health personnel, mental health programs with a community approach, access to assisted fertilization programs, and the generation of collaborative alliances between health services, civil society organizations, and the LGBTIQ+ community. We conclude that the health system is a crucial space from which to enable guarantees for the exercise of rights and overcome internalized stigma.

## 1. Introduction

The heterocisnormative Judeo-Christian model (HJCM), present in Latin America [1,2,3,4], promotes ideologies regarding what constitutes a “normal” family, gender roles, and normative social milestones to be accomplished, denaturalizing sexual and gender diversity and validating discrimination and subordination of LGBTIQ+ people. The HJCM sustains, at a sociocultural level, homo/lesbo/bi/transphobia and discrimination [1,2,3,4]. Stigma, prejudice, and discrimination create a hostile and stressful social environment for LGBTIQ+ people that can cause mental health problems, expectations of rejection, hiding/camouflaging (performing as a hetero cisgender person), internalized homo/lesbo/bi/transphobia, and ameliorative coping processes [5].

Heterosexual-cisgender parenting is considered the foundational nucleus of the family [6,7]. Parenting is confused with fecundity, associating LGBTIQ+ people with sterility, arguing that a couple formed by LGBTIQ+ people cannot reproduce “naturally” and, therefore, could not experience maternity/paternity [8,9,10]. The same criterion is not applied when couples formed by cisgender heterosexual people (CHP) are infertile and perform procedures of adoption and assisted reproduction techniques [8,11,12].

This discrimination is not based on the (in)existence of a blood relationship between children and parents but on the sexual orientation/gender identity of the people who exercise the parental role, contrary to recognizing human rights [11]. Policies, laws, and governmental programs have perpetuated violations of LGBTIQ+ people’s rights [13], for instance, restraining the access of LGBTIQ+ people to reproduction techniques provided by public health systems.

There is a criminalization/sanction of LGBTIQ+ people that is (re)produced in micro-social environments as the family of origin. LGBTIQ+ people experience a system of homo/lesbo/bi/transphobia and discrimination in intimate spaces: 52.7% of the LGBT population has suffered some type of direct discrimination based on their sexual orientation/gender identity, including 58.1% of trans people, 57.4% of lesbians, 49.1% of gay men, and 48.8% of bisexuals [14]. This discrimination promotes the development of internalized stigma, and the incorporation/incarnation of prejudice, which results in LGBTIQ+ people expressing rejection towards themselves and other people in the LGBTIQ+ community [15,16].

Health services constitute a potential support network regarding how maternity/paternity will be faced from diversity perspectives. A person with few support networks is more likely to adapt to the heterocisnormative model, renouncing the desire to experience maternity/paternity [17,18]. However, public health systems in Latin America are weak; they have little funding/personnel available. Higher quality health services are associated with the private sector [19], the payment sector, which LGBTIQ+ people cannot access due to their socioeconomic conditions. Community health approaches would be beneficial, given their history of concrete links with social movements and the defense of the rights of vulnerable groups [20].

Regarding mental health services, although in Latin America there has been a declared interest in generating guidelines for mental health professionals, there is still a lag in the attention to their particular needs, considering their specific problems [15,16,21,22,23]. Mental health interventions should be oriented to guarantee the rights of the LGBTIQ+ community [15,16,21,22,23].

Internationally studies on LGBTIQ+ families have focused mainly on LGB people [24]; the study of TIQ+-headed families is still an emerging area [24,25]. Some authors argue that TIQ+-headed families research should be done through qualitative methodologies that allow an in-depth understanding of the phenomenon [25,26,27].

In this article, we present part of the results of research conducted from a feminist perspective. We seek to make visible the experience of parenting of LGBTIQ+ people and their particularities. We intend to nourish the local-situated analysis with knowledge generated in Latin American environments based on a rights approach. We interviewed people from Chile, Colombia, and Mexico, because we wanted to contrast parenting experiences lived under different sociocultural and legal frameworks. 

When this study started, Chile had no gender identity law (the law was approved during the period of the production of information of this study), same-sex marriage was prohibited by law, and adoption was not legally allowed for LGBTIQ+ parents—even though some same-sex couples could adopt children under exceptional circumstances. Colombia has same-sex marriage law, but adoption and gender identity changes are not legally recognized. In some states, Mexico has same-sex marriage law; in other states, adoption is allowed, and gender identity law was being discussed and approved during the period of production of information of this research. We considered those differences to build a sociostructural sampling that allowed us to access different experiences of LGBTIQ+ people and their points of view about the possibilities and obstacles regarding parenting.

We have reported the results generated in this research in several papers. Specifically, in this article, we will present the effects of internalized stigma on the exercising of parenthood from LGBT people’s perspective and particular demands that the LGBTIQ+ community raises in health care regarding access and inclusion.

## 2. Materials and Methods

Participants were 21 LGBT people and key informants, belonging to academia, psychotherapy, politics, and diversity activism, over 18 years old, from Chile (16), Mexico (4), and Colombia (1); the participants were people between 21 and 57 years of age, with a mean age of 37.19 and a standard deviation of 10.03. We show other sociodemographic information in Figure 1. We recruited key informants through their academic or social profiles. Participants were invited through social media open calls and selected according to the theoretical sample to fulfill the sociostructural categories delineated, drawn from a sociostructural sampling [28,29], as shown in Figure 1.

We used the biographical method through open-ended interviews [30]. The biographical approach allows the production of information through the moments and points of inflection in people’s lives in the paradigmatic border space between the individual and the social structure. The biographical method enables the study of problems through a limited number of cases that address the issue, encompassing macro- and micro-sociological issues that sustain power systems in the production, organization, and maintenance of inequalities [30]. We are connected to a sociohistorical context that inhabits us and appears in our discourses [30]. We applied two interviews, one focused on building the intersectional loom; the second one focused on LGBTIQ+ people’s parenting.

Since there were open interviews, the script was flexible and could be changed across the interview, guided by some main focus, such as identifying the participant’s social identities and their categories, differences perceived between some rights’ guarantees, an evaluation of camouflage effect performed (trying to be perceived as a hetero cisgender person, to guarantee access to some rights), obstacles, emotions, help received, aids needed, and differences perceived in the experience of parenting while being an LGBTIQ+ person.

As researchers, we conducted analyses and contrast until the information was saturated [31]; likewise, the participants were able to discuss research findings at different moments of the process, which had the objective of modulating the processes of (re)presentation of people [32,33] and legitimizing the interpretation of their discourses through triangulation processes. We applied biographical interviews focused on identities. Then, we built the intersectional loom with each participant. Later, we did a biographical interview focused on the research topic, when identities and the intersectional loom could be modified if necessary, according to the participant’s discourse. We transcribed and analyzed the interviews, identifying emerging thematic axes and identities. We did new interviews in some cases to fill the gaps of information. Then, we prepared an analysis grid, including theoretical counterpoints. We made diagrams that were shown to the participants to check their agreement to the conclusions made by the research team. Finally, we prepared some information dissemination formats (as a scientific paper) which were shown to the participants. Then, we interviewed new participants, following the logical sequence that the sociostructural sampling suggested, making this process circular. We offer this process in Figure 2.

This research was approved by the authorized Scientific Ethical Committee of Universidad Católica del Norte (bioethics committee resolution 039/2017) in compliance with legal safeguards. The interviews were coded according to the initials of each participant or with a self-assigned pseudonym to guarantee the confidentiality of participants. We will destroy interview records and their transcriptions after five years, as the Chilean legal framework mandates. Table 1 shows the codes corresponding only to the interviews reproduced in this article, which are part of the total number of interviews carried out. Since we wanted to analyze the representations that LGBT people had about maternity/paternity, we interviewed people who have children and those who have not. We expected LGBTIQ+ people to tell us why they did not want or cannot have children because we thought that information could also be relevant regarding the obstacles that can appear to accessing parenting while being an LGBT person.

## 3. Results

### 3.1. Internalized Stigma

There is a criminalization/sanctification of sex and gender diversity and a homo/lesbo/bi/transphobia system that LGBTIQ+ people experience as discrimination in their intimate and broader social environments. That discrimination could generate an internalization of prejudice. LGBTIQ+ people can manifest rejection towards themselves and other members of the LGBTIQ+ community [5,15,16]; this rejection has different expressions, and we will review those related to parenthood.

#### 3.1.1. (Im)Possibility of Thinking of Oneself as a Mother/Father

This internalized stigma impacts their self-perception as potential mothers/fathers and their ability to think of themselves as people who can build a family:
Even in diverse people, it is sentenced that “we cannot be parents because we cannot”. So, they just try to accept themselves, but they cannot see that another LGBTIQ+ person can go further and form a family.(J.O., personal communication, 23 April 2019)
All those ghosts appear, that I am not going to be enough because: “Maricón” [derogatory nickname for gay men] or that I am not going to be enough because “I do not have the same characteristics of …”, even I am saying: “It does not correspond”, “I do not have to think that” … I feel that all these situations, that upbringing and the whole model and also it is supported by a social issue, more or less imposing and inquisitive, has come to condition a lot my perception regarding fatherhood, me being a father [Emphasizes] and especially thinking me being a father with my homosexual partner [Emphasizes] … When the conversation has come out, precisely those fears or those limitations that have to do with my own construction, as a subject … It terrifies me, it scares me not to be enough … those structures that have been installed survive, survive, those structures are maintained for many years and there are people of diversity who can never get rid of them or who fight all their lives with them. I think it is essential to mention it, fear is perhaps the very subjective category, but that transcends or involves everything.(Maximiliano, personal communication, 28 May 2020)

#### 3.1.2. Fear of Violating Children’s Rights

Due to the internalized stigma that some LGBTIQ+ people have [5,17,34], judgments against themselves, feelings of degradation, disadvantage or the idea that oneself is not who one should be, or the perception of being evaluated or seen as inappropriate by others appear [35]. Stigma and prejudice become informal mechanisms of social control of gender identities/sexual orientations that move away from the heterocisnorm [34]. Those stigmas and prejudices are incorporated by LGBTIQ+ people, leading them to question their right to access/experience maternity/paternity [17,18], which results in the belief that, by exercising parenthood, the LGBTIQ+ person would be violating the rights of children by depriving them of having a “normal” family:
Above all, I believe that there are reasons that may be linked to the fear of being judged as a bad father or bad mother for being gay, bisexual, lesbian, or trans, for being a violator of the child’s rights.(M.R., personal communication, 31 May 2019)
What will happen to the child? Because you put yourself in that position [beind discriminated] before and, in one way or another, you play the martyr, you say “Ok, I don’t care!”. But exposing a child to that situation also makes you question yourself … [because you are] putting him in dangerous situations, painful situations.(Maximiliano, personal communication, 28 May 2020)

#### 3.1.3. Fear of Passing on the Stigma to Children

LGBTIQ+ people constantly deal with discrimination and homo/lesbo/bi/transphobia [2,3,5,15,16]. Such discrimination impacts social and institutional environments, leading them to experience emotions such as fear of discrimination and anguish due to transfer of the effects of such discrimination to their children:
The fear was a subject that we talked about a lot, a lot of time, and we always wanted to be mothers. But we were afraid about what others might say … When you go out, the issue is a fear one has, more than for oneself, but for the children. That they will be rejected too … So, that is the main obstacle: one’s fear.(Saau, personal communication, 3 February 2020)
I would feel observed … I would feel more concerned about my son in the case of … what would they say to him … what would I say to him if they say something to him and I cannot defend him, or I cannot explain him … I do not know, there are things that I could not answer; maybe it will not happen, but, if it happens, how will I react, how will I protect him. Even if he is little… he will always have to carry that social stigma of having gay parents … [people would think] “And since they are homosexual, the child is also homosexual” … if they say things to me, it does not matter to me, I am an adult, but if they say something to, in this case, is my son. If they hurt him, it will hurt me; it will hurt me. In that case, how do I protect him? … Although maybe there is not so much discrimination… but there is a rejection… or a social reproach, people have their opinion, there are critical people, and they criticize the whole family…. Because they are gay parents, the child is also to blame.(Mau, personal communication, 7 March 2018)

#### 3.1.4. Fear of Presenting an LGBTIQ+ Partner

There are different ways LGBTIQ+ people can be parents, including having children in a heterosexual relationship [17,18]. In the case that the mother/father experiences an LGBTIQ+ relationship after the birth of their children, due to the internalized stigma, there is a conflict/tension related to disclosing their sexual orientation/gender identity to their children:
My partner has two children, with two different women, and he is with me now. Still, he cannot talk about it with his family because they are very homophobic … other people don’t know about our relationship either. Still, we have a relationship [do his children know about your relationship?] We haven’t told them, but I think they understand it implicitly. Many people have a relationship with a person of the same sex, who has children, but they don’t day that it is taboo because it is not well seen in the society in which we live.(Aron, personal communication, 7 March 2018)
The conversation was difficult for me, not for him [his son] … I put a lot of effort into it, I made it very complicated, but not for him. According to my perception, for him, it was easy. It took me a long time to tell him. Although he knew it was obvious, I had to speak it and tell him directly, “I am with a same-sex partner”. It was so hard I did not think it was going to be that difficult. Until I could tell him, and he told me, “I know, I support you, I love you… ok, calm down, dad. I love you as you are, and you have to be calm, and I will support you”. It was like taking a tremendous weight off me. I kind of started sleeping a lot better that day. Because I had a weight, a thing … a burden that did not let me be calm. Until I talked about it openly with him, I no longer have those same restrictions from the moment I spoke. I hug my partner more I am closer to my partner. As we discussed and my son does not mind, I would have liked, having assumed what it was before, to be more calmly participating in my child’s upbringing, as in parenthood.(Sebastian, personal communication, 10 January 2020)

#### 3.1.5. Greater Difficulties in the Experience of Parenthood for Intersex/Trans People

Regarding the situation of trans persons, some studies show that levels of sexual prejudice towards the trans population are higher than towards gays, lesbians, or bisexuals [14,15,16,36,37]. The most significant social difficulties faced by trans and intersex people also appear concerning the experience of parenthood:
In the case of intersex or trans people, I feel that they are even more misunderstood than a person of diversity in general. A person of sexual diversity who is also intersex or trans has a more significant burden before society and much more discrimination. it is more difficult for them.(Sebastian, personal communication, 10 January 2020)
First, for someone who is trans… We talk about it with my friends as they are gay or bi[sexual]. First, the fact of assuming that your partner is trans. There is a matter of transphobia … First, it is that barrier. The second is that the mere fact of being trans and publicly being trans … they are more violated. People do not assume their name, the name that they have. They do not endorse, perhaps the way of dressing, “I know that his genitalia is female. Why does he feel like a man?” And so. And that is added to the fact that they are in a relationship with someone and have children: “No, this person is a deviant, degenerate person”. They end up being much more violated than a cisgender person. They would be excluded in all social aspects. At least my friend, she, being on the street, feels discriminated. And it is sad because she wants to be the way she feels.(Ivan, personal communication, 1 January 2020)

#### 3.1.6. Generate Psychoeducation on LGBTIQ+ Parenting as a Strategy

Participants highlight the relevance of generating socioeducational strategies that disseminate experiences on LGBTIQ+ parenting and truthful scientific information in an appropriate and easy to understand way, which could reduce the prejudices that people have towards LGBTIQ+ parenting and can help LGBTIQ+ people to buffer/resist internalized stigma:
To end or reduce discrimination, the stigmas that exist need to show parenthood experiences from diversity. To show that nothing happens, that the world is not ending, it is not terrible. That boys and girls can be just as happy with heterosexual parenthood as with LGBTIQ+ people’s parenting, to show that, the reality … So that can be normalized, so it is not something strange or wrong.(Sebastian, personal communication, 10 January 2020)
Being gay or being a lesbian or being trans does not negatively affect your son or daughter’s sexual orientation and gender identity; but rather, what has a negative impact is the prejudice or discrimination concerning those sexual orientations and gender identity. Eventually, the risk lies in stigmatization due to negative attitudes towards this type of family. We would say, by society in general. If it did not exist, there would be no danger. So, all the studies show that there is no danger in the child’s psychological well-being. There may be problems in integration, but not since they live with two mothers or two fathers, but rather the stigma associated with it, but exercised by the general population.(J.B., personal communication, 7 January 2019)

### 3.2. Demands on the Health Care System

The LGBTIQ+ population faces socioeconomic and labor precariousness that commonly begins with difficulties in remaining in the educational system [15,16,38]. This precariousness leads to challenges in consulting, and fear of seeking care and accompaniment from, a public health professional [15,16].

#### 3.2.1. Free Health Services

Participants argue that it is vital to guarantee free access to health care so that socioeconomic gaps are not an impediment for LGBTIQ+ people to exercise their right to health:
We need that the State establishes support plans, understanding that parenting is not the same as being a parent of sexual diversity. There should be support from the State to provide support to these people … guidance, psychological support, accompaniment.(Sebastian, personal communication, 10 January 2020)
It is a class issue. It is much easier when you are in a high social and economic segment; I mean, you can enter into the parenting dynamic. … there is a play on words we use “It is very different to be gay than to be a fag”, the gay man is a homosexual who has money. Based on that social and economic position, you have freedom. On the other hand, the fag who has no money is screwed.(Maximiliano, personal communication, 28 May 2020)

#### 3.2.2. Gender Identity Perspective and Inclusive Treatment Provided by Health Personnel

The HJCM is (re)produced by people in micro-social environments; it permeates our discourses on the distribution of symbolic and material social goods and about people who should or should not have them guaranteed [39,40]. Given this, it is relevant to train health personnel to raise awareness of the treatment and discourses contrary to universal access to health care as a right:
For us trans people, if we go to a medical service, either Government or private health service, which is instead government health service, because no job pays you to have private medical assistance. We run into the discriminatory issue, from not wanting to attend you, not understanding or not empathizing a little bit with your gender identity, making it visible that you are not the person you project.(S.T., personal communication, 7 April 2020)
Training professionals is essential. Midwives in the clinics are not prepared to take care of bi or lesbian girls … one of the main focuses is educating the nurses … they could take the opportunity to give information that does not appear in the books. It is dangerous, it is dangerous that we are so ignorant … yes, they discriminate, but they discriminate out of ignorance. It should not be that simple. They would not have to ask stupid questions … I do not even want to think about what a 16-year-old girl thinks, who wants to know a little more, and has to ask the midwife; or if a woman has to report that her girlfriend hit her, how does she do it?(Alicia, personal communication, 11 February 2020)

#### 3.2.3. Mental Health Programs with a Community Approach

In Latin America, as a result of the binary HJCM of gender roles/relationships, the management of the processes at the psychotherapeutic level is developed under a lack of resources that allow an effective solution to the problem [15,16], which impacts the (im)possibility of exercising parental roles from the sex-affective/gender diversity perspective:
The first time a support group was done with people of diversity, it was in the framework of alcohol and drugs health program that focused on diversity, and that was the only way the program could be accepted, and then they carried it out.(J.O., personal communication, 23 April 2019)
In her case [her partner], there is this kind of perform as heterosexual, for example, at work. We have reflected on it, had to ask for advice, we have done therapy to consider these things like: “Hey if the children are going to school, you have to assume it [being lesbian]”.(Saau, personal communication, 3 February 2020)

The participant reveals that support programs for LGBTIQ+ people must be camouflaged as other programs to have state funding to serve the LGBTIQ+ population. In this case, under the umbrella of an addiction treatment program, they could meet the demands of LGBTIQ+ people.
The positive effect that access to individual therapy has on LGBTIQ+ people could be crucial. Group therapy or support groups where information and experiences can be shared become relevant too. Interventions with a community focus would have an additional positive effect in terms of promoting self-management and the formation of stable intersectoral support networks, where nurturing exchanges between LGBTIQ+ people could be generated: “For diverse paternities and maternities, I tell you, I haven’t seen it. I always see aids in general… not as something specific”.(Ivan, personal communication, 1 January 2020)
LGBTIQ+ parents take their children to psychologists, and they also actively participate in these sessions … Group workshops, where the experience could be shared, where doubts could be demystified. I don’t know, whatever may appear as a doubt within this maternity or paternity. To generate links with the educational places where they have their children. To open the subject, not to close it, because it should be normalized.(C.V., personal communication, 16 December 2019)

#### 3.2.4. Guarantee Access to Assisted Fertilization Programs

There is a continuum regarding the possibility of deciding to become a mother/father while being an LGBTIQ+ person, in which voluntariness and prior decisions regarding the process (for example: deciding to have a known or unknown sperm donor; saving money to pay for in vitro fertilization; stopping hormone therapy to be able to conceive, in the case of trans people) allow some LGBTIQ+ people to evaluate whether or not they want to have children and under what circumstances, considering the legal, economic, and social limitations to which they are exposed [17,18]:
The fertilization through FONASA [Chile’s public health system], which is now open, has unlimited quotas that were limited before. Even so, they require you to be married to a man. So that’s how it was, and I remember that it was very controversial because single women and women who are married to other women are not eligible for the benefit that FONASA gives, supposedly, to the entire population, so it was a bit difficult.(Na, personal communication, 17 May 2019)
First, the main obstacle is that here in Chile, we do not have … How do you say this? The access … that you could, let’s say, get pregnant with sperm … you have to do it totally through the private health system, which is very expensive. And, on the other hand, there are countries like Argentina, where everyone can access assisted fertilization, where the State subsidizes you. If you have any problem, regardless of whether you may have fertility problems or not … In my case, the most challenging thing was that the economic cost … it would be my second treatment. We are not going to continue like this either … it is already a lot of money, so no. Spending so much… That is always there. We talked about this problem, how many times to try, within the possibilities.(Saau, personal communication, 3 February 2020)

#### 3.2.5. Generate Collaborative Alliances with a Community Approach between Health Services, Civil Society, and LGBTIQ+ Organizations

Mental health interventions for LGBTIQ+ people in Latin America still have a predominant focus on disclosing sex-affective orientation or gender identity [16]. There is ample room for the growth of other areas of intervention, such as those related to the experience of parenthood, through collaborative alliances with a community approach that could be established between health services, civil society, and LGBTIQ+ groups:
There should be an institution of its own, which, if it is going to be governed by the State or, in this case, by the Municipality, should be a house of diversity exclusively or people of sexual diversity … focused on problems… current problems of sexual diversity, which are: adoption, being thrown out of the house. These aggressions are becoming more and more notorious … there is a lack of support.(J.O., personal communication, 23 April 2019)
[Interventions could be made in] spaces of greater informality to work with neighborhood associations, workshops, and things like that. They may be able to enter other areas that are not traditional. I say this because I feel that way, you cannot wait ten, fifteen, twenty years for cultural changes to occur regarding what the children are being taught; I believe that intervention has to be more concrete and more profound from today, from now on. In that sense, precisely incorporating education in non-traditional spaces is decisive to generate change.(Maximiliano, personal communication, 28 May 2020)

## 4. Discussion

In this article, we expose the effects of internalized stigma on the experience of parenting from the perspective of LGBTIQ+ people and present particular demands that the LGBTIQ+ community raises regarding current problems in health care and interventions to support them in this specific issue.

We found that internalized stigma impacts the parenting of LGBTIQ+ people in five ways: the impossibility of thinking of oneself as a mother/father, fear of violating children’s rights by wanting to experience parenting, fear of passing on the stigma of being an LGBTIQ+ person to their children, fear of introducing their LGBTIQ+ partner when the children do not know the sexual orientation or gender identity of the mother/father, the greater discrimination that trans and intersex people suffer in this area. We also describe the generation of psychoeducation strategies on LGBTIQ+ parenting as a tactic of resistance/buffering against prejudice and internalized stigma.

About the (im)possibility of thinking of oneself as a mother/father, there is a homo/lesbo/bi/transphobia system that LGBTIQ+ people experience as discrimination in their intimate and broader social environments. Internalized stigma is the internalization of those prejudices. Internalized stigma impacts the perception of LGBTIQ+ people about their possibility of becoming a mother/father and makes them doubt their ability/capacity to form a family.

LGBTIQ+ people incorporate stigmas and prejudices, including one that establishes that children have the right to have a “normal” family that LGBTIQ+ people, supposedly, cannot provide, so the fear of violating children’s rights by exercising parenting appears.

Since LGBTIQ+ people have suffered the effects of homo/lesbo/bi/transphobia, they are susceptible to and concerned about the possibility of passing the stigma to their children. LGBTIQ+ people generate a fear that can even make them dismiss the idea of becoming a mother/father, as has been told by our participants who have no children.

If an LGBTIQ+ person has children conceived in a heterosexual relationship, being in a same-sex relationship or changing gender identity becomes a tough conversation that parents are afraid to have with their children. Participants highlight the fear of presenting an LGBTIQ+ partner when their children do not know their sexual orientation/gender identity.

The trans population faces a more significant risk of being discriminated against than cisgender people and other sexual minorities such as gay, lesbian, and bisexual people; intersex people have the poor visibility of their conflicts as an added obstacle. Those more significant social difficulties that trans and intersex people face also appear regarding the experience of parenting.

LGBTIQ+ people are active actresses/actors in the social field, so they stand against the reproduction of social inequalities. One strategy that LGBTIQ+ people developed to resist marginalization is to generate instances of psychoeducation on LGBTIQ+ parenting. Those instances should have proper support to survive the test of time; participants stand up to the relevance of being supported by local or national public health system/institutions/public policies.

Regarding the demands directed towards the health sector that could facilitate the exercise of the rights of LGBTIQ+ people, gaps in health care were identified, which must be filled to guarantee universal access to health care, a gender identity perspective, and inclusive treatment provided by health personnel, mental health programs with a community approach, access to assisted fertilization programs, and the generation of collaborative alliances with a community approach between health services, civil society organizations, and the LGBTIQ+ community.

Due to difficulties in staying in the school system, the LGBTIQ+ population commonly faces socioeconomic and labor precariousness that results in problems accessing private health services, so it is urgent to provide free physical and mental health services guaranteed by laws and public policies. 

We are subjects who inhabit a sociohistorical context that, at the same time, inhabits us and appears in our discourses. Social discourses include assumptions about symbolic and material social goods and people who should or should not have access to them and certain social positions, as the parental role. Given the HJCM that prevails in Latin America, participants argue that it is relevant to train health personnel about gender identity perspectives and inclusive treatments to visualize the effects of reproducing discourses contrary to the exercise of the right of LGBTIQ+ people.

The HJCM establishes gender roles and models of relations, pathologizing non-binary identities, transitions, and the exercise of gender identities or expressions that escape the heterocisnormative model. Given this, it becomes crucial to ensure mental health programs with a community approach, to facilitate the access of LGBTIQ+ people who might not be in contact with health institutions but can be related to other community organizations.

LGBTIQ+ people access parenting through various routes, in which voluntariness, decision making, and prior planning have a preponderant weight regarding the conditions in which parenting will occur, including deciding to get pregnant through a donor, considering whether the donor will be a known or unknown person; planning a savings program that allows them to pay for an assisted fertilization; or stopping taking hormone therapy to allow a pregnancy in trans people, to name a few. Sociostructural situations intersect these ways of accessing parenthood, such as socioeconomic level, that can enable LGBTIQ+ people to pay for assisted fertilization even though participants highlight that there should be access to assisted fertilization programs guaranteed by law/public policies.

LGBTIQ+ people consider that their ability to parenting is inadequate due to the reproduction of the discourse that the heteronormative family and the hegemonic model of being a mother/father are the only correct ways to experience parenting. Having incorporated the dominant position, people (re)produce intersecting social inequalities and marginalize themselves from the rights exercised by members of the dominant class. Participants argue that generating collaborative alliances with a community approach between health services, civil society, and LGBTIQ+ organizations could help them get information, know their rights, and exercise them, demanding the fulfillment of their rights that are violated.

Our study aimed to understand the parenting experiences of LGBTIQ+ people. We found similar results to previous studies in Latin American contexts regarding gay and lesbian parenting [6,7,9,17,18]. We are pleased to provide information on trans parenting since this area of research is emerging in Latin America [34] and around the world [24,25,27].

We postulate that interventions aimed at LGBTIQ+ people from the health system should have a community and intersectoral approach, with intense work with the communities that will be the beneficiaries of programs [19]. The LGBTIQ+ community should be considered from diagnosis of the system’s current state to interventions and modifications that should be made in the system and health programs.

The recursive process applied in the production and analysis of the data allowed us to present results that have been validated both by the triangulation carried out within the research team and by the triangulation carried out with the participants.

Although evidence shows that the development of children is not negatively affected by the experience of growing up in a family that includes an LGBTIQ+ person, the existence of alleged adverse effects related to this type of family is still prevalent. We consider establishing social/legal differences based on gender identity and sexual orientation of people who exercise parental roles as unfair and contrary to the exercise of rights.

Regarding the implications of this research work, we believe that we provide relevant information about modifications that must be implemented in health systems. It is imperative to move towards the construction of legal frameworks that recognize diverse families’ existence and give them protection and a feeling of equality before the law that the LGBTIQ+ community demands. We need to correct unequal social, economic, and political conditions to build a world where social justice is a reality for everyone and not a pending task.

Regarding the study’s limitations, we consider the impossibility of interviewing people from rural environments and with a low level of schooling. Given that the recruitment took place through social networks, the sample was limited to people who used those information technologies.

## 5. Conclusions

This article shows the effects of internalized stigma on the experience of parenting from LGBT people’s perspectives and particular demands that the LGBTIQ+ community raises in health care regarding access and inclusion.

We believe that the health system is an institution that structures and (re)produces violence and inequalities through its practices; therefore, it is a crucial space from which to enable guarantees for the exercise of rights with more participatory, community-based strategies focused on collectives.

The ways of configuring parental arrangements, how people access parenting, the ideas about being a mother/father, and how parents establish their relationships with their children, among others, are being rethought in same-sex and opposite-sex couples. Then, it is contrary to the rights perspective to establish differences based on the sexual orientation/gender identity of the members of the marital block [17,18].

However, we have not yet been able to build a legal framework that recognizes the existence of diverse families and gives them protection and equality before the law. The defense of human rights is confronted with unequal social relations, economic conditions, and political structures. In this sense, the task of making social justice a reality for all mains a pending one.

## Figures and Tables

**Figure 1 ijerph-18-05373-f001:**
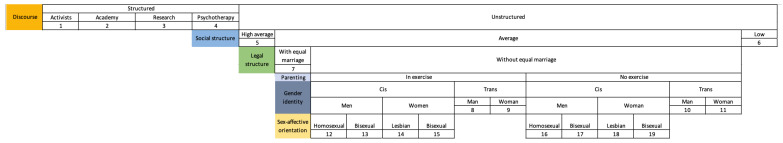
Sociostructural sampling.

**Figure 2 ijerph-18-05373-f002:**
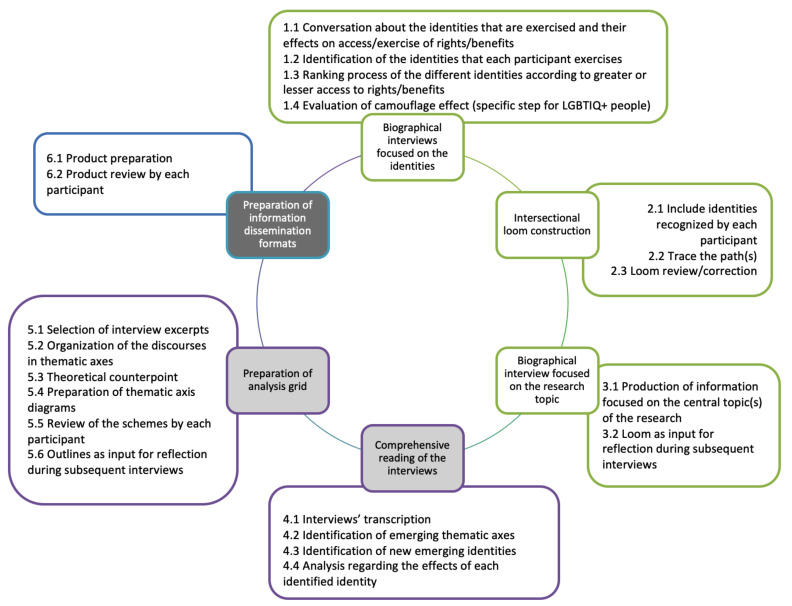
Research process.

**Table 1 ijerph-18-05373-t001:** Coding of participants.

Characterization Data	Participant Pseudonym
Trans man, activist, 21 years old, Chile, zero children	J.O.
Gay male, researcher, 47 years old, Colombia, zero children	M.R.
Lesbian woman, psychologist, 37 years old, Chile, two children	Saau
Gay man, make-up artist, 30 years old, Chile, two children	Aaron
Bisexual man, self-employed, 34 years old, Chile, one child	Sebastian
Trans woman, activist, 34 years old, Mexico, zero children	S.T.
Bisexual woman, psychotherapist, 34 years old, Chile, zero children	C.V.
Lesbian woman, physician, 24 years old, Chile, zero children	Na
Bisexual man, student, 27 years old, Chile, zero children	Ivan
Bisexual man, teacher, 33 years old, Chile, zero children	Maximiliano
Gay man, psychologist, 33 years old, Chile, zero children	Mau
Gay man, researcher, 47 years old, Chile, zero children	J.B.
Bisexual woman, self-employed, 35 years old, Chile, one child	Alicia

## Data Availability

The data presented in this study are available on request from the corresponding author. The data are not publicly available due to restrictions included in the informed consent signed by the study participants.

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
