# Peer review of "The Impact of Internalized Stigma on LGBT Parenting and the Importance of Health Care Structures: A Qualitative Study"

_ijerph, 2021, doi:10.3390/ijerph18105373_

Round 1
Reviewer 1 Report
This is an important study and, of course, a topic that remains timely for many nations. I very much enjoyed reading this and can see the corroboration of the data with other discussions on this topic.
I would urge the authors to consider developing their discussion, though, further, as it is very brief and does not do justice to the findings. More analytical discussion is needed and in relation to other sources.
Secondly, the authors could draw on more sources available around LGBTQ+ parenting - especially about the tensions in relation to internalised stigma.
Reviewer 2 Report
First of all, thank the authors for their work entitled " LGBTIQ+ people's maternities/paternities: internalized stigma and demands on health care, a qualitative study”, which reflects in a very explicit way the need to continue to make visible and fight for LGBTIQ+ equality. I consider that it has an interesting approach for publication in Int. J. Environ. Res. But, there are some questions of form that should be taken into account prior to consider this article for publication.
I enclose my suggestions for consideration by the authors.
- I recommend that the authors reflect in the methodology who was the head of recruitment in social networks.
- The authors should introduce the approval code of the bioethics committee in the methodology.
- The authors should reflect if the interviews were eliminated after transcription.
- I imagine that the interviews followed a script, the authors should publish it.
- The results would have more value if the authors reflected two statements of the interviewees per item treated.
- The discussion must be reformulated, the authors must reel off point by point, comparing each point with previous results, reflecting and reflecting what social intervention they would take to improve the aspect.
- The conclusions do not respond clearly to the objective.
Reviewer 3 Report
Thank you for the opportunity to review the paper “LGBTIQ+ people's maternities/paternities: internalized stigma and demands on health care, a qualitative study”. Although I think it has some potential for IJERPH’s readers, the following comments apply:
- Title does not reflect participants: it mentions LGBTIQ+ people, but only LGBT people participated. I suggest that authors change the title to provide more accuracy.
- Also, I would suggest authors to revise the title according to what was really done. Something in this line would be more accurate: the impact of internalized stigma on LGBT parenting and the importance of health care structures: a qualitative study.
- Line 16: “We identified gaps in health care” I would suggest rephrasing to “We identified gaps in health care perceptions”.
- If the study was conducted with participants from Chile, Mexico and Colombia, cultural specificities of these culture should be empathized both in the title and in the abstract, and also in the introduction and discussion sections.
- Lines 25-29: I would suggest the inclusion of references about the Sexual Minority Theory Stress (Meyer) and how they apply to Latin American reality.
- Lines 30-35: I would suggest uniformization of nomenclature: parenthood vs parenting vs paternity vs parental role… it gets confusing.
- Line 39 requires further explanations: Violations of LGBTIQ+ people´s rights are perpetuated through policies, laws and governmental programs [12].
- Lines 41-48: I suggest the re-writing of this paragraph. The ideas are hard to follow. Also, the word “violence” accurately fits the results of reference 13?
- Lines 49-53: what are the specificities of health care systems in Latin America?
- Lines 55-58: Why is this paragraph devoted to mental health professionals? This seems unexpected since it’s beyond the Scopus of the paper. Unless authors want to focus on both physical and mental health services, and this should be clarified.
- Lines 59-63: this is unfunded. Please search for several examples around the world regarding lgbt-headed families and their social representations, and add some recent references.
- Lines 64-68: send this to the discussion section.
- Please clearly present the research goals, as they are dubious and emersed with assumptions: “we will present the effects that internalized stigma has on the exercise of parenthood from diversity and certain demands that the LGBTIQ+ community raises in health care regarding this specific issue” it’s not accurate. You measured peoples’ opinions or representations, right? Also, health care in which dimensions? Access? Quality of care? Inclusion? Etc.
- Line 80 – More information on recruitment of participants is necessary. What does “according to their accessibility” even mean?
- Line 81 - The biographical method needs to be further explained and updated references.
- Line 90 – Figure 2 should not replace information regarding the explanation of the research process in the text. Please provide a brief view of this process and then complement it with figure 2.
- If Participants were 21 LGBT people and key informants how come table 1 only provides information regarding 9 people? Please provide clarification.
- Also, if this is a study about LGBT people's maternities/paternities, how come table 1 informs about 6 participants with zero children? Please provide clarification.
- The materials and methods section lacks a “materials” paragraph with detailed information. Sociodemographic information, interview script, online webpage?
- I have a trouble accepting that “(Im)possibility of thinking of oneself as a mother/father” and “Fear of violating children’s rights” is a direct measure of peoples’ internalized stigma.
- Line 186: Authors should refrain from expressing their beliefs when reporting results.
- Line 193: Gender identity?
- It’s not clear to my what authors mean by “health demands”. Please provide clarification.
- Discussion needs further explanations. Please provide comparison with other previously conducted research. Please provide specific social characteristics that apply to Latin American context that may have interfered with results. Please provide a “limitations” section. Please provide an “implications” section.
Round 2
Reviewer 3 Report
Thank you very much for addressing all the issues I raised in my previous review. Overall, the paper has very much improved.
Best wishes.